# Signal Conditioning Stage in S-Band Communication Subsystem for CubeSat Applications

**Joel A. Castillo** [1],* [ID]**, Jorge Flores-Troncoso** [1] [ID]**, Rigoberto Jáuregui** [2]**, Jorge Simón** [3] **and José L. Alvarez-Flores** [1]

1   Engineering and Applied Technology, Autonomous University of Zacatecas, Zacatecas 98000, Mexico; jflorest@uaz.edu.mx (J.F.-T.); alvarez_jose@ucol.mx (J.L.A.-F.)
2   Mexican Navy Institute for the Research and Development, Mexican Navy, Veracruz 95269, Mexico; jauregui@inidetam.edu.mx
3   Catedras CONACYT, Autonomous University of Zacatecas, Zacatecas 98000, Mexico; jsimonro@conacyt.mx
*   Correspondence: 26701212@uaz.edu.mx; Tel.: +52-1-492-107-2990

**Abstract:** This work presents the design, performance evaluation, manufacture, and characterization of an RF front-end signal conditioning chain on a substrate that achieves the best performance at S-band frequencies and complies with the dimensions of the international standard for CubeSat-type nanosatellites. In this development, the signal conditioning chains were carried out on the high-frequency substrates RO4350B, CuClad 250, and RT/duroid 5880, considering scattering parameters in a small-signal regime. Concerning the power output, after the filtering and amplifying stages, the conditioning chain delivered 2 watts at 2.25 GHz. Moreover, up to 40 dB gain was achieved, and a good impedance matching at −20 dB for both input and output ports was observed. The numerical simulations and experimental results showed that an RO4350B substrate allows the smallest design dimensions, and these comply with the dimensions of the CubeSat standard. The manufactured RF front-end signal conditioning chain on RO4350B requires an area of 95 mm$^2$, and it is ready to be used in a proof-of-concept space mission in a CubeSat.

**Keywords:** S-band; signal conditioning; CubeSat standard

## 1. Introduction

With the recent growth of the global space industry, modern small satellites play a relevant role in many aspects of the space economy. The arrival of commercially available, high-volume, and low-cost microelectronics has led to miniature satellites called nanosatellites. These small satellites are being used in low Earth orbit (LEO) for earth observations, communications, and interplanetary missions, among others. There is an international design standard for a nanosatellite of a unit, the 1U CubeSat. It is a 10 cm cube with a mass of less than 1.33 kg. With these size restrictions, RF and microwave circuitry for communication subsystems should be as small as possible [1–4].

An RF front-end of a CubeSat communication subsystem is an essential functional element and has two critical parts for transmission and reception chains, that is, amplification and filtering stages. Most satellite applications work at frequencies above 1 GHz; for example, for space research in Mexico, the frequency band from 2.2 to 2.3 GHz (S-band range) is assigned [5]. Specialized substrates are necessary to design and fabricate high-frequency power amplifiers on a printed circuit board (PCB). Substrates such as RO4350b, CuClad 250, RT/duroid 5880, RO4003C, RO3003, and TC350 meet requirements such as permittivity and loss tangent. Substrate properties define the width and length of the microstrip lines, which are the basis of traditional distributed microwave components and circuits. These properties limit the usage of any substrate characteristic because of CubeSat dimension limits. At the amplification stage, the maximum design limitation are the input and output matching ports, which force the optimization of the substrate properties to obtain the desired dimensions [6–8]. According to the method used in [9] for the downlink

budget (from CubeSat to earth station), for an LEO at 600 Km, a transmission output power of 33 dBm is enough to receive telemetry information, even an image at the ground station.

As an example of this kind of signal conditioning chain, the authors in [10] presented an S-band microwave transmitter (STX) built using commercial off-the-shelf (COTS) parts, with a maximum power of 30 dBm for a 1U CubeSat. Additionally, COTS parts are used for signal conditioning stages with a 33 dBm transmission power for nanosatellite applications in the S-band [11,12]. In [13,14], some transmitters in the S-band are described, but all of them are COTS. The authors in these works do not perform the calculations of the matching networks; they use amplifiers given by the manufacturers.

The drawback of [10–14] is the high cost; sometimes, the desired device is not available. With this work, we intend that lectors will be able to design and fabricate their own RF amplifier, and it complies with all the requirements as much as possible.

This work aims to design, evaluate, manufacture, and characterize RF front-end signal conditioning chains on a PCB substrate that present the best performance at the S-band and an output power of up to 2 W in the small-signal regime and comply with CubeSat standard dimensions. The proposed procedure includes the design and numerical comparisons of the performance of RF front-end signal conditioning chains on RO4350B, CuClad 250, and RT/Duroid 5880® substrates that meet requirements such as permittivity and loss tangent at high frequencies. In all cases, the printed microstrip lines are used for designing the microwave circuitry, including amplifiers' matching networks.

The article is organized as follows. In Section 2, some particular properties of the substrates used in this approach are described. It also gives an overview of the signal flow, including the passive and active devices used in this work for filtering and power amplifying, and the design methodology of an RF amplifier is presented. A dimensional comparison is performed in Section 3, where amplifier matching networks, using microstrip technology, are designed and tested for several substrates, and a theoretical comparison is performed between some substrate designs using ADS software simulations. Additionally, characterizations are shown experimentally, including power and gain performance, and the obtained results are discussed. Finally, the conclusions are provided in Section 4.

## 2. Signal Conditioning Chain

### 2.1. RF Substrates

The laminates used in this approach, RO4350 [15], CuClad 250 [16], and RT/Duroid 5880 [17], have specific properties in high-frequency applications, with permittivity being the main property. The substrates were chosen because of the availability and the characteristics for this application. Permittivity is the main factor that restricts the dimensions of a microwave circuit. Table 1 shows some of the specification parameters for the used substrates, such as permittivity, one of the most critical parameters for the designs presented in this work.

**Table 1.** Comparison of three well-known RF substrates showing the properties of interest.

| Parameter | Substrate | | |
|---|---|---|---|
| | RO4350B | CuClad 250 | RT/Duroid 5880 |
| Permittivity | $3.48 \pm 0.05$ | 2.4–2.6 | $2.2 \pm 0.02$ |
| Loss tangent | 0.0037 | 0.0018 | 0.009 |
| Thermal coefficient | +50 | −170 | −125 |
| Thermal conductivity | 0.69 | 0.254 | 0.2 |

### 2.2. RF Front-End Signal Conditioning Chain Architecture

A signal conditioning stage, also known as the amplification and filtering stage, is the last stage before the signal is sent to the antenna to be transmitted. The architecture design presented in this work is well known. Signal conditioning chains are commonly used in biomedical, satellite, cellular, and other applications, which, depending on the application, may include a different type of filter or a number of amplifiers required. In this article,

they are composed of a band-pass filter, a driver amplifier, and a power amplifier. The block diagram in Figure 1 shows a typical signal conditioning stage (blue block), where it can be observed that there are some matching networks between stages.

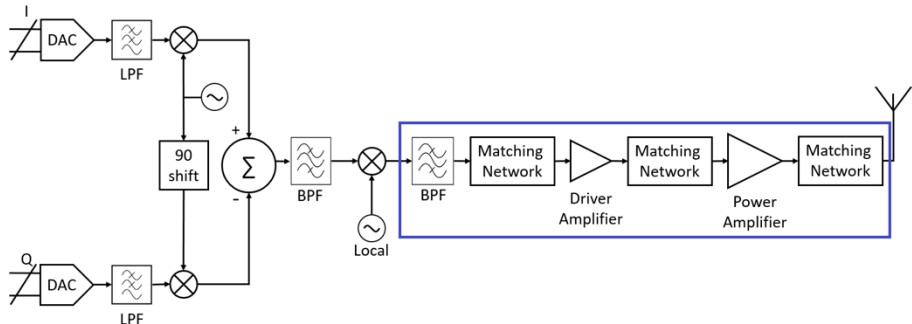

**Figure 1.** Typical signal conditioning stage.

The band-pass filter allows only specific frequencies located within a given bandwidth pass through and attenuates those outside this range. At the output of a mixer, as shown in Figure 1, it is widespread to have multiple signal levels at nondesired frequencies, including harmonics and intermodulation products, which have to be suppressed for the correct operation of the following amplification blocks, avoiding saturation and unwanted conversions. Usually, N-order filters are designed based on passive components, such as resistors, capacitors, and inductors, but problems appear when the theory is put into practice since in most cases, the values for these components result far from commercial values. On the other hand, there are several kinds of microstrip filters, but designing microstrip filters implies a compromise with dimensions and efficiency. Nowadays, we can find integrated circuit filters with outstanding performance, which simplifies the size and whose performance is similar to that of a high-order filter. In this work, for all designs, the DEA162495BT-1289A1 filter was selected. It is a multilayer band-pass filter, which works from 2200 to 2700 MHz and whose dimensions are as small as 1.6 mm × 1.8 mm [18].

The driver amplifier, also known as the preamplifier, is responsible for delivering power to drive the power amplifier into a desired operation (i.e., the preamplifier supplies power to the chain to reach the desired output power level). The QPA6489A MMIC microwave amplifier from Qorvo was selected as driver amplifier at a frequency range of DC to 3500 MHz, small-signal gain of 19 dB, and 19 dBm output power, which fulfills the power requirements at the input of the power amplifier [19].

The power amplifier is located at the end of the stage and is responsible for delivering the required power to the antenna, in this case, 33 dBm. The amplifier was designed from the MMZ25332BT1 transistor, which provides the desired power from 1.5 to 2.8 GHz with a gain of up to 25 dB. The main advantage of MMZ25332BT1 is the 5 V bias control, which meets the requirements for a CubeSat energy power subsystem [20].

*2.3. Design Procedure*

This subsection presents the signal conditioning stage design procedure under the small-signal regime, where the amplifier stage design is prioritized. The design methodology is as follows:

Step 1. Data performance

The first step in designing an amplifier, regardless of the power operation, is finding the data performance, usually provided by the manufacturer as a mathematical model or particular parameters. It is possible to have a linear or nonlinear model, where the first one involves the scattering parameters (S-parameters). By using specialized microwave software, such as Advanced Design System (ADS) from Keysight® (Santa Rosa, CA, USA), we can use S-parameters to find any linear magnitude, such as impedance and reflection coefficient. In order to design these power levels, typically small-signal conditions are

used. Therefore, in this work, the S-parameter design method was used. For high-level power amplifiers (tens or hundreds of watts), a nonlinear model is needed, besides other design techniques, such as load-pull. However, the analysis and design of such a nonlinear model are beyond the scope of this work because for this medium power application, a small-signal model is valid for modeling the behavior of the amplifier used in this work.

Step 2. Amplifier stability

A critical consideration in an amplifier design is stability or its resistance to oscillation. Having S-parameters at the design frequency, the first condition to satisfy is the stability factor, and it is expressed as [6,7]

$$k = \frac{1 - |S_{11}|^2 - |S_{22}|^2 + |\Delta|^2}{2|S_{12}S_{21}|}, \tag{1}$$

where $\Delta$ is the measure of stability given by

$$\Delta = S_{11}S_{22} - S_{12}S_{21}. \tag{2}$$

Stability occurs when the following condition is satisfied:

$$|\Delta| > 1 \ \& \ k > 1. \tag{3}$$

Step 3. Optimal impedance

The best performance of an amplifier occurs when the matching networks are a simultaneous conjugate match. For this, it is necessary to find the optimum impedance "seen" by the amplifier at the input and output ports, which are given by Equations (4) and (5) for input and output, respectively:

$$Z_{MS} = Z_0 \left( \frac{1 + \Gamma_{MS}}{1 - \Gamma_{MS}} \right), \tag{4}$$

$$Z_{ML} = Z_0 \left( \frac{1 + \Gamma_{ML}}{1 - \Gamma_{ML}} \right), \tag{5}$$

where $Z_0$ is the system impedance, usually 50 $\Omega$, while $\Gamma_{MS}$ and $\Gamma_{ML}$ are the reflection coefficients at the input and output and written as

$$\Gamma_{MS} = \frac{B_1 \pm \sqrt{B_1^2 - 4|C_1|^2}}{2C_1}, \tag{6}$$

$$\Gamma_{ML} = \frac{B_2 \pm \sqrt{B_2^2 - 4|C_2|^2}}{2C_2}. \tag{7}$$

The variables $B_1$, $B_2$, $C_1$ and $C_2$ are defined as

$$B_1 = 1 + |S_{11}|^2 - |S_{22}|^2 - |\Delta|^2, \tag{8}$$

$$B_2 = 1 + |S_{22}|^2 - |S_{11}|^2 - |\Delta|^2, \tag{9}$$

$$C_1 = S_{11}^2 - \Delta S_{22}^*, \tag{10}$$

$$C_2 = S_{22}^2 - \Delta S_{11}^*. \tag{11}$$

Step 4. Matching networks

There are several techniques for matching impedance, depending on the design requirements. L-networks using lumped components such as resistors, capacitors, and in-

ductors are well known. A common way to represent a matching with L-networks is using the reactance and susceptance as shown in Figure 2, which represent lumped components, but for circuits at frequencies above 1 GHz, it is more appropriate to use microstrip lines since most of the values required from lumped components are not available. Microstrip lines in the substrate are commonly used to replace lumped components.

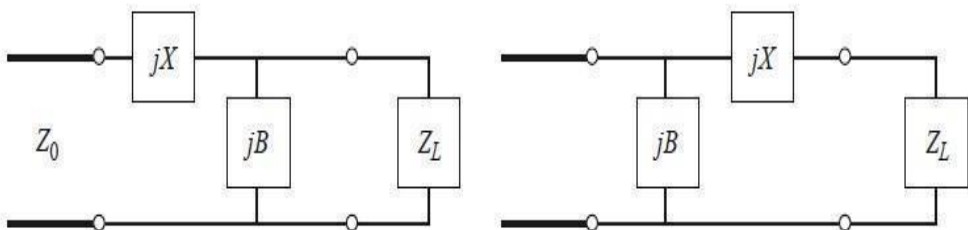

**Figure 2.** L-section matching networks.

　　A single stub is a simple technique for matching microwave circuits. A stub is a microstrip line connected to the transmission path to change the microstrip line reactance. In single-stub tuning, the two adjustable parameters are the distance, *d*, from the load to the stub position and the reactance value provided by the stub. For single-stub matching, there are two cases: shunt stub and series stub. For the shunt-stub case, the basic idea is to select *d* so that the admittance, Y, "seen" looking into the line at distance *d* from the load is of the form $Y_0 + jB$. Then the stub susceptance is chosen as -*jB*, resulting in a matched condition.

　　The fundamental values to be determined are *d*, already discussed, and *l*, the length of the stub, defined by Equations (12) and (13), respectively.

$$d = \begin{cases} \dfrac{\lambda}{2\pi} tan^{-1}(t) & for \ t \geq 0 \\ \dfrac{\lambda}{2\pi}\left(\pi + tan^{-1}(t)\right) & for \ t < 0 \end{cases}, \tag{12}$$

$$l = -\frac{\lambda}{2\pi} tan^{-1}\left(\frac{B}{G}\right). \tag{13}$$

　　Suppose that the length *l* is negative, then $\frac{\lambda}{2}$ can be added to give a positive result. $\lambda$ represents the speed of the wave in the material, and *G*, *t*, and *B* are given by

$$G = \frac{1}{Z_0}, \tag{14}$$

$$t = \frac{X_L \pm \sqrt{R_L\left[(Z_0 - R_L)^2 + X_L^2\right]/Z_0}}{R_L - Z_0} \quad for \ R_L \neq Z_0, \tag{15}$$

$$B = \frac{R_L^2 t - (Z_0 - X_L t)(X_L + Z_0 t)}{Z_0\left[R_L^2 + (X_L + Z_0 t)^2\right]}. \tag{16}$$

$Z_0$ is the impedance of the system, and $X_L$ and $R_L$ are the reactance and resistance of the load obtained from the impedance calculated in Equations (5) and (6) for the input and output.

### 2.4. Model of Signal Conditioning Stage Design

　　Figure 3 draws a general diagram in ADS Keysight® software for the signal conditioning stage. TermG1 and TermG2 represent the ports of a network analyzer, with 50 Ω of characteristic impedance. After TermG1, an S-parameter block of the band-pass filter is presented, followed by a coupling capacitor, which blocks DC signals; TL1, TL2, TL3, and TL4 are part of the matching networks of the driver amplifier, while TL11, TL12, TL13, and TL14 correspond to the power amplifier. After that, there is one more coupling

capacitor. Driver and power amplifiers are represented by their S-parameter blocks named like their corresponding part numbers.

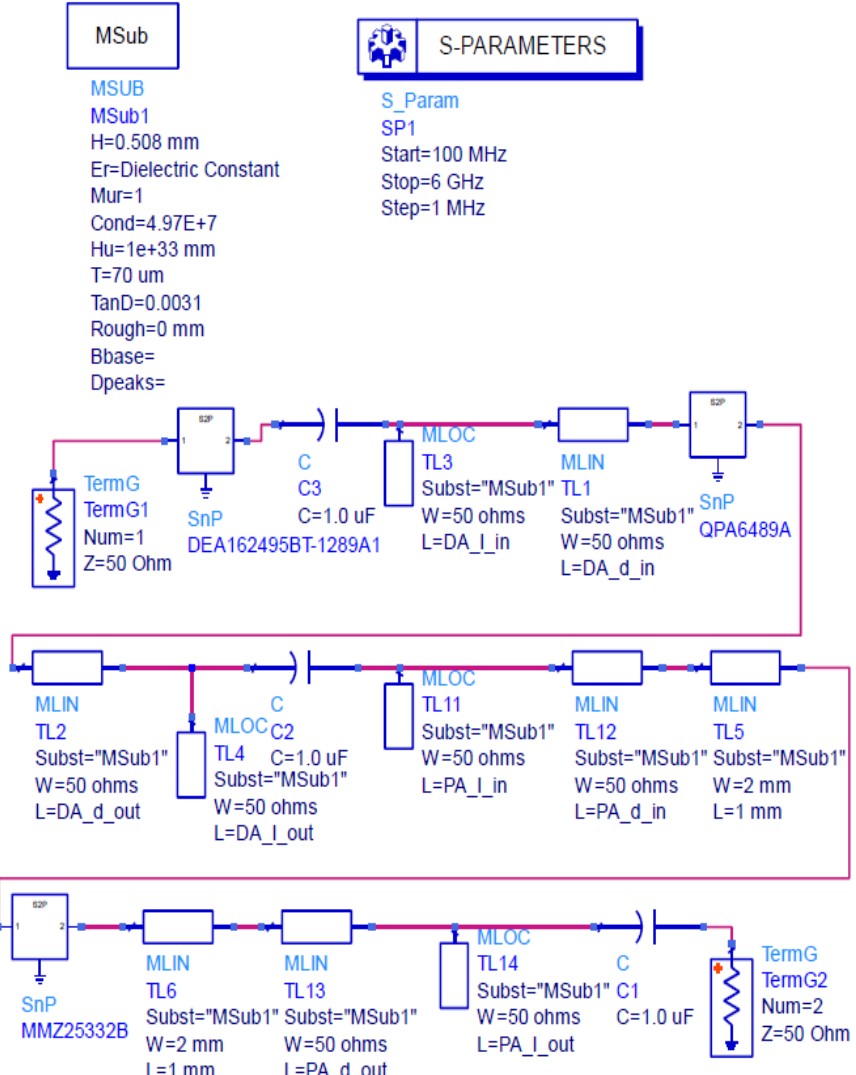

**Figure 3.** Model and simulation of signal conditioning stage designed for each substrate carried out in ADS software.

*2.5. System Simulation*

An S-parameter simulation was carried out in ADS for each of the proposed substrates, substituting substrate parameters from Table 1 in MSUB, as shown in Figure 3, and a sweep in frequency from 100 MHz to 6 GHz with 1 MHz steps was executed.

Unconditional Stability

S-parameter simulation allows for knowing whether the chain will be unconditionally or conditionally stable; when Rollet's conditions presented in Equation (3) are met, the signal conditioning stage is unconditionally stable. These conditions can be visualized, adding StabFact for k and StabMeas for Δ to the S-parameter simulation in ADS.

*2.6. Fabrication*

The PCB was designed using a specialized software called Altium Designer, and its fabrication was carried out with a milling machine and gold plating to prevent oxidation. Figure 4 shows the prototype manufactured on RO4350B, which hardly fits on a CubeSat

PCB. This prototype measures 9.5 cm per side, with cutouts in the corners to allow the placement of the poles that hold the CubeSat structure, complying with the norms of the standard given by [3,21]. This prototype weighs 30 g, making it highly viable for a CubeSat, where weight and size limitations are crucial.

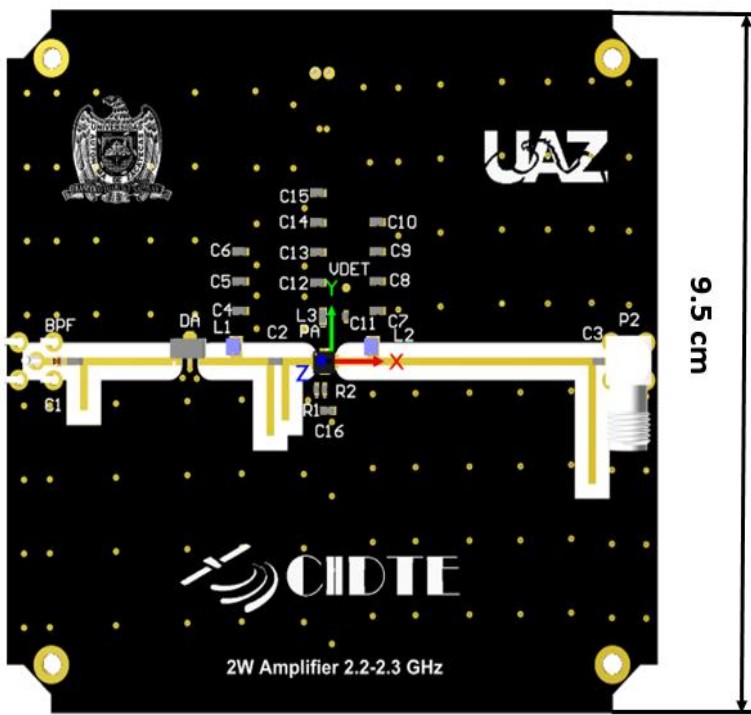

**Figure 4.** Final approach of the signal conditioning stage.

### 2.7. Characterization

Once the PCB was manufactured and assembled, the characterization of the parameters of interest was performed using high-performance measurements.

The S-parameter characterization needs a vector network analyzer (VNA), where the gain magnitude of the $S_{21}$ parameter for a desired sweep of frequencies can be found. Figure 5 shows a commonly used test bench for measuring microwave amplifiers. An N5222A VNA from Keysight Technologies is used, calibrated at a −20 dBm power level.

Attenuators are required at the output of the amplifier to protect the equipment from damage due to overload. A 5 V power supply biased the amplifiers. In this case, only transmission parameters were used to determine the gain and power characteristics, previously considered the attenuators' effects in the calibration process.

To measure the peak power, and because of the high gain of the chain, a calibration sweep driving power was set from −30 to 0 dBm using a VNA.

An arbitrary waveform generator (AWG), M8190A, from Keysight was configured to introduce typical satellite modulated signals to the signal conditioning chain and evaluate the performance against possible real applications. The injected modulated signals were amplitude-shift keying (ASK), quadrature phase shift keying (QPSK), quadrature amplitude modulation (16-QAM), and amplitude and phase-shift keying (16-APSK). All modulated signals worked at a carrier frequency of 2.25 GHz and a symbol rate of 500 Mbaud. An essential parameter for amplifiers is the power-added efficiency (PAE), and it is set as high as possible for satellite applications. For that, a 2.25 GHz sinusoidal wave from −21 to −4.8 dBm was applied to the device under test (DUT) input, and then we observed the output power (POUT) through a signal analyzer. Therefore, for each power value, the PAE was calculated as

$$PAE\ (\%) = \frac{P_{OUT} - P_{IN}}{P_{DC}} \times 100. \tag{17}$$

Moreover, to analyze the signal distortion and harmonic effects at the power amplifier output, a 2.25 GHz sinusoidal tone was applied to its input. In the first instance, the original output signal was observed, and then it was compared with the resulting output when using a DEA162495BT-1289A1 band-pass filter.

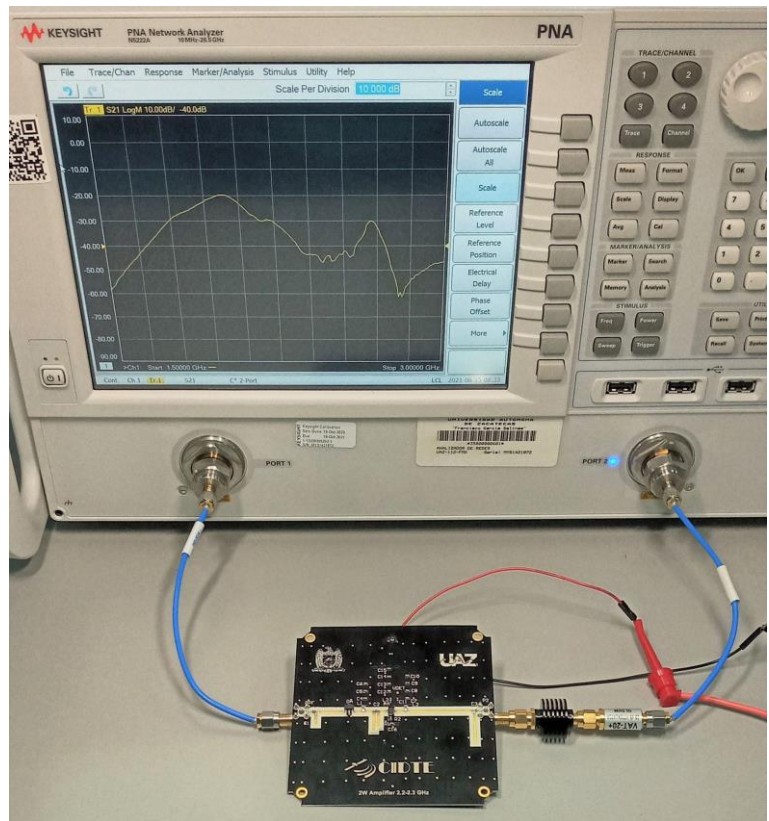

**Figure 5.** Setup for measuring the transmission parameter ($S_{21}$) by using a vector network analyzer (VNA).

## 3. Results and Discussion

### 3.1. Matching Networks Comparison

Table 2 shows the length (*l*) of the stub and the distance (*d*) from the stub to the input and output ports of the driver amplifier (DA) and the power amplifier (PA), respectively, and the width of the microstrip lines to ensure a 50 Ω environment.

**Table 2.** Dimensions for microstrip lines for matching networks considering three substrates.

| Measurement | Substrate | | |
|---|---|---|---|
| | **RO4350B** | **CuClad 250** | **RT/Duroid 5880** |
| Width for 50 Ω | 1.08 mm | 1.37 mm | 1.5 mm |
| DA_d_in | 14.28 mm | 16.32 mm | 17.16 mm |
| DA_l_in | 6.62 mm | 7.57 mm | 7.96 mm |
| DA_d_out | 10.49 mm | 11.99 mm | 12.61 mm |
| DA_l_out | 10.08 mm | 11.52 mm | 12.11 mm |
| PA_d_in | 3.77 mm | 4.31 mm | 4.53 mm |
| PA_l_in | 7.99 mm | 9.14 mm | 9.61 mm |
| PA_d_out | 37.52 mm | 42.88 mm | 45.1 mm |
| PA_l_out | 16.56 mm | 18.93 mm | 19.91 mm |

As shown in Table 2, dimensions vary depending on the substrate characteristics. As observed, the suitable dimensions are obtained using RO4350B, and the largest using

RT/duroid 5880. Depending on the design requirements, the appropriate properties need to be selected. In this particular case, with the nature of the CubeSat limitations, size optimization is desired.

As mentioned in Section 2.6, the signal conditioning stage hardly fits on a CubeSat PCB, so it used RO4350B values for matching networks. If we used CuClad 250 or RT/duroid 5880 lengths, they would hardly fit as the RO4350B design achieved it.

### 3.2. Simulation Results

Figures 6–8 show a comparison of the basic parameters, such as input and output reflection parameters, $S_{11}$ and $S_{22}$, and forward transmission parameter, $S_{21}$. A similar behavior is observed for each substrate, obtaining a relatively acceptable impedance matching lower than −20 dB at the input and output ports and achieving a gain of up to 40 dB at a frequency of interest of 2.25 GHz. For those mentioned above, the selected substrate for fabrication is mainly due to the physical design dimensions (length and width).

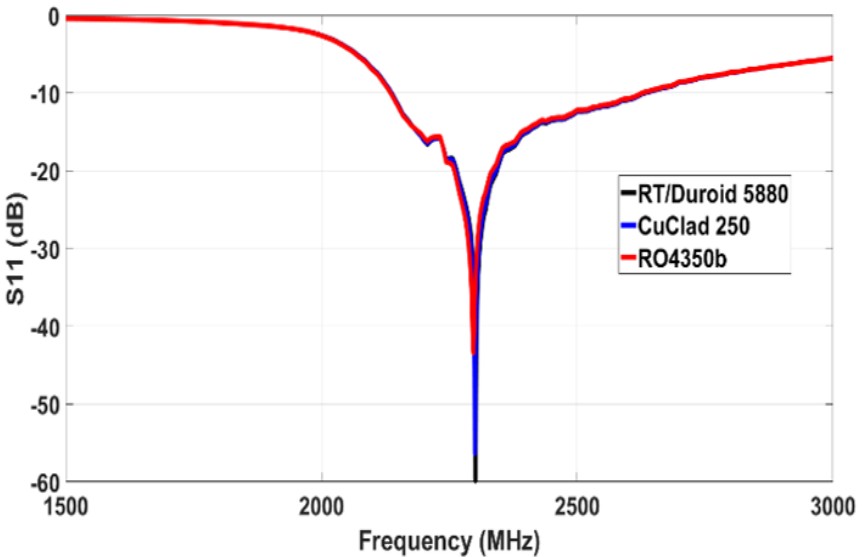

**Figure 6.** Input reflection parameter ($S_{11}$) simulation for the selected RF substrates centered at 2.25 GHz.

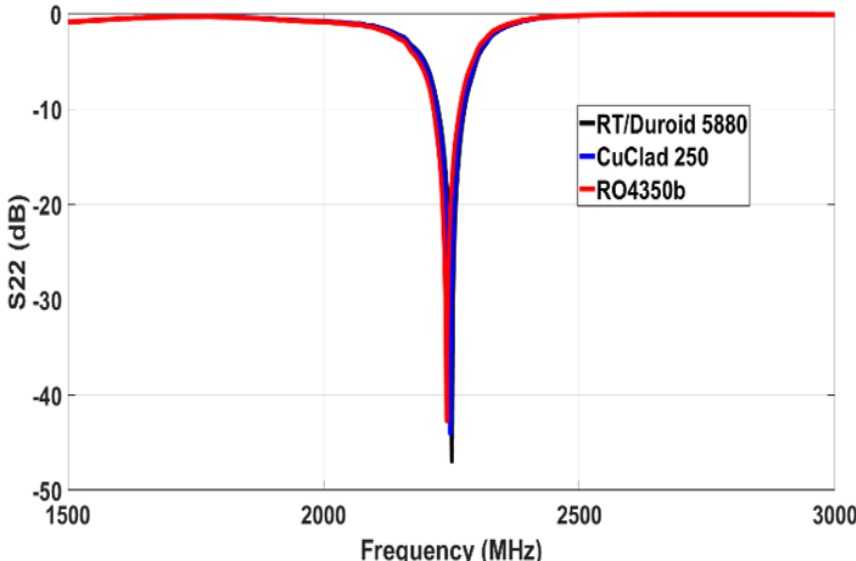

**Figure 7.** Output reflection parameter ($S_{22}$) simulation for the selected RF substrates centered at 2.25 GHz.

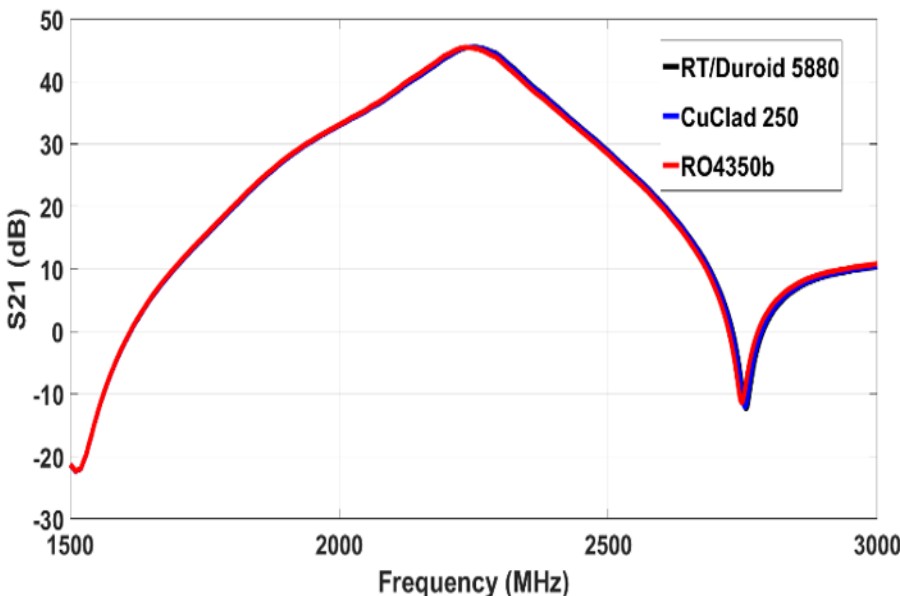

**Figure 8.** Forward transmission parameter (S$_{21}$) simulation for the selected RF substrates centered at 2.25 GHz.

Figure 9 shows a comparison of stability factor (k) at 2.25 GHz k > 1 for each of the substrates; therefore, the first condition is complied.

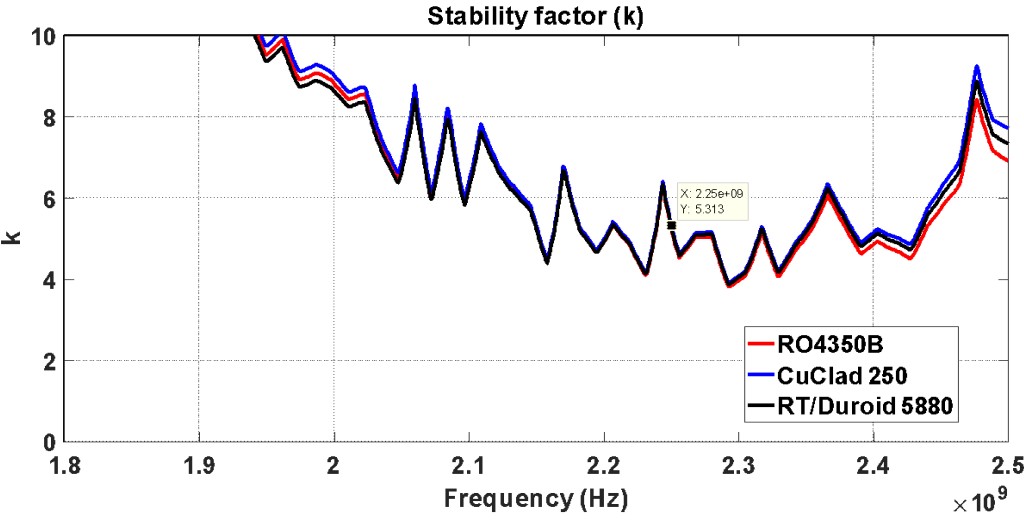

**Figure 9.** Stability factor (k) simulation for the selected RF substrates at 2.25 GHz.

The stability measure ($|\Delta|$) for the selected RF substrates is shown in Figure 10. In this case, only a RO4350B chain meets with $|\Delta| < 1$, so only this design will be unconditionally stable.

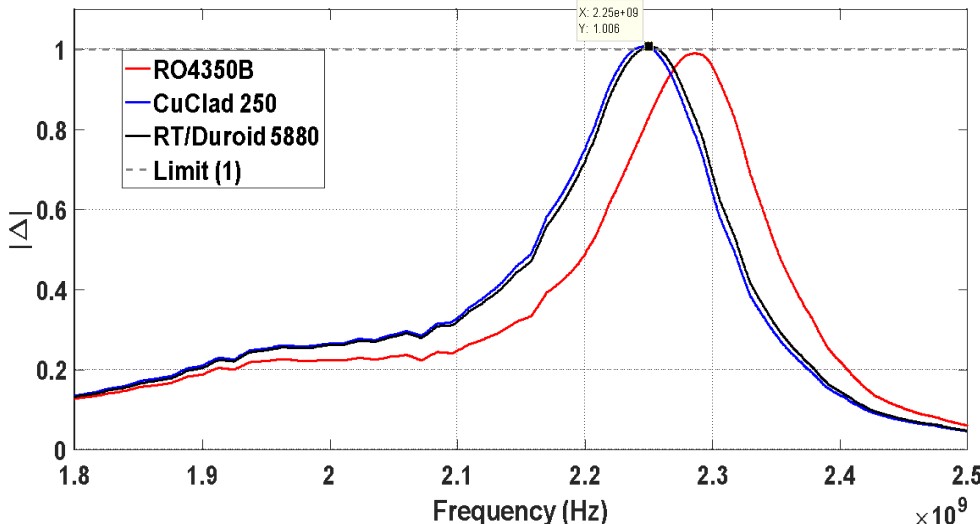

**Figure 10.** Stability measure ($|\Delta|$) simulation for the selected RF substrates at 2.25 GHz.

### 3.3. Experimental Results

Figure 11 shows a comparison between measured and simulated gains with a suitable correlation in terms of bandwidth. The conditioning signal stage offers acceptable performance from 1900 to 2330 MHz.

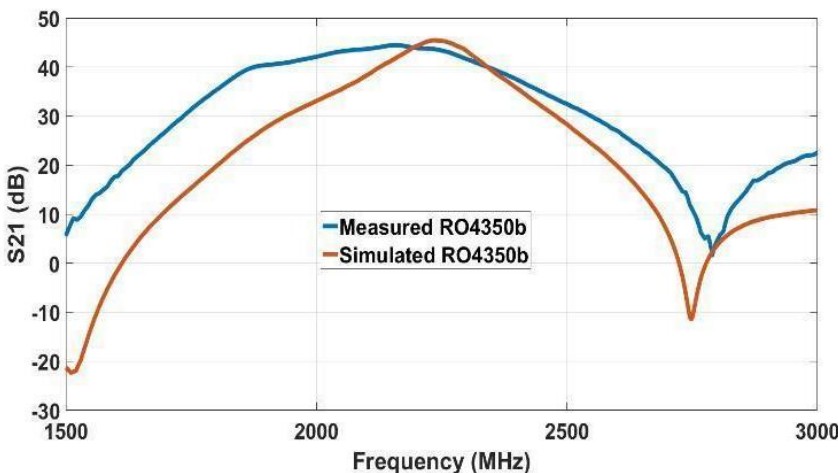

**Figure 11.** Small-signal transmission characteristics of the simulated and measured results for a RO4350B substrate centered at 2.25 GHz.

Figure 12 depicts the gain compression as a function of the input power and the output power delivered along the power sweep at a single frequency of 2.25 GHz. As observed, the gain is higher than 40 dB for small-signal conditions and about 38 dBm for the 33 dBm output power needed. Once the compression level is crossed, the gain falls rapidly because the output power saturates and is not longer.

Figure 13 shows the performance of the signal conditioning stage with a different kind of modulation, which is good, given the types of modulations that were used. The main lobe of each modulation is affected because the bandwidth of the signal is greater than what the amplifier can work. This can be improved by reducing the bandwidth of the modulations.

Finally, in Figure 14, a plot of PAE vs. POUT is shown. It is worth mentioning that PAE increases as the signal conditioning stage becomes saturated. In this case, the signal conditioning chain reaches 45% at its maximum output power of 33 dBm.

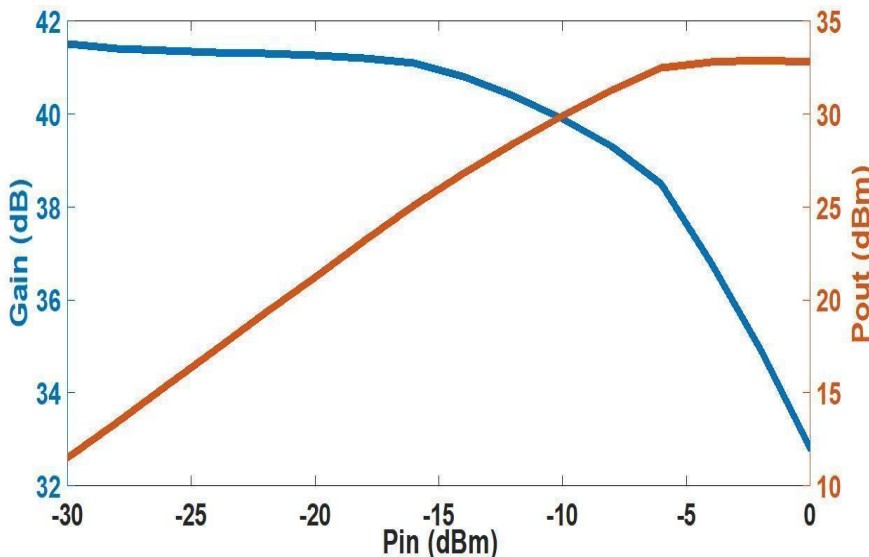

**Figure 12.** Power and gain characteristics of the conditioning block at 2.25 GHz.

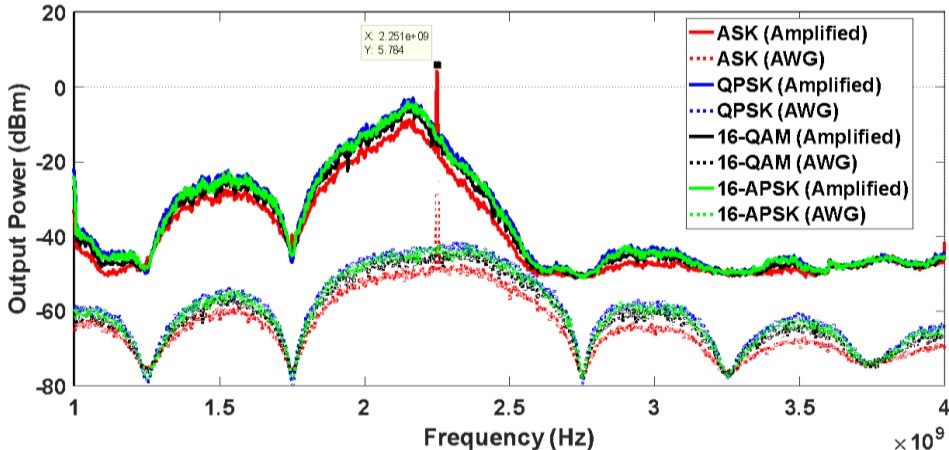

**Figure 13.** Performance of the signal condition chain with different modulations.

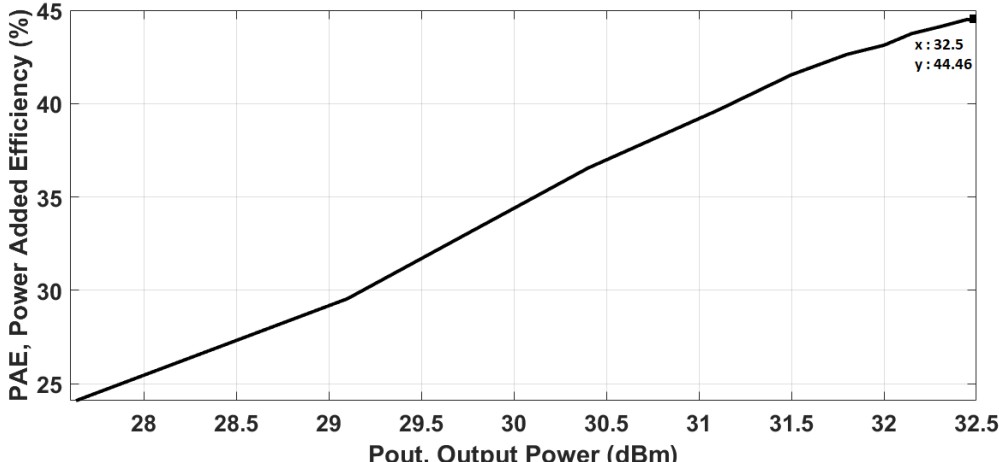

**Figure 14.** PAE of the signal conditioning stage.

Figure 15 shows a comparison of signal at the power amplifier output without a band-pass filter in blue and with filter in red. It shows how the filter performs harmonic suppression, but it affects output power, decreasing it up to 4 dB.

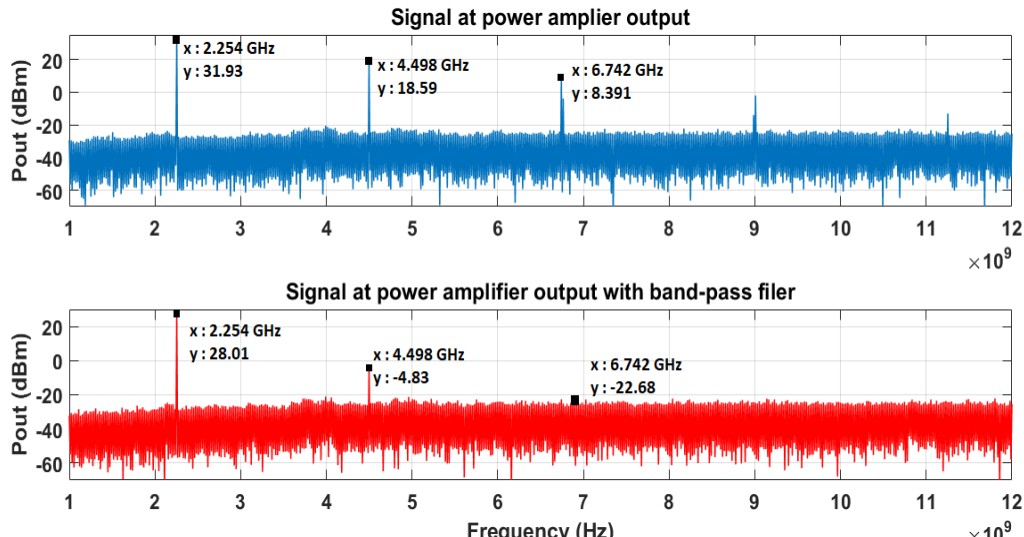

**Figure 15.** Comparison of signal at power amplifier output with and without band-pass filter.

## 4. Conclusions

In this article, a comparison of a signal conditioning stage designed on three different substrate technologies—RO4350B, CuClad 250, and RT/duroid 5880—was carried out. As a result, we showed a similar performance on simulation but varying physical dimensions. However, only the RO4350B substrate complied with the CubeSat dimension requirements and Rollet's conditions for stability. Then, a prototype for the signal conditioning stage was designed and manufactured using RO4350B, resulting in a 2-watt chain operating at 2.25 GHz. This prototype achieved a gain of up to 40 dB with good impedance matching of less than $-20$ dB at the input and output ports. A power-added efficiency of 45% was achieved at its maximum output power and good performance against real-world modulations. In addition to a large bandwidth of about 400 MHz, this prototype is able to work in the frequency range of 1900 to 2330 MHz. Furthermore, due to similar values of copper cladding and substrate thickness, it is shown that the size of the matching networks is inversely proportional to the permittivity of the substrate.

**Author Contributions:** Writing—original draft preparation, J.A.C.; writing—review and editing, J.F.-T. and J.S.; methodology and formal analysis, R.J.; investigation and validation, J.L.A.-F. All authors have read and agreed to the published version of the manuscript.

**Funding:** The authors would like to thank the Consejo Nacional de Ciencia y Tecnologia (CONACYT) for the support given to Joel A. Castillo with scholarship number 469812 and CVU 637098 to carry out his postgraduate studies.

**Data Availability Statement:** The data presented in this study are available on request from the corresponding author.

**Conflicts of Interest:** The authors declare no conflict of interest.

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
