# Peer review of "Signal Conditioning Stage in S-Band Communication Subsystem for CubeSat Applications"

_electronics, doi:10.3390/electronics10141627_

Round 1
Reviewer 1 Report
Manuscript: electronics-1229023 The manuscript titled 'Signal conditioning stage in S-band communication subsystem for CubeSat applications' by authors Joel A. Castillo et al present results on the RF Front-End signal conditioning chain on different substrates for a proof-of-concept space mission in a CubeSat. I have following observations: 1// Among the substrates discussed, for example line 45 *Substrates as RO4350b, CuClad 250, RT/Duroid 5880, RO4003C, RO3003, and TC350*, the reason to chose RO4350B, CuClad 250, and RT/Duroid 915880 alone (line 91) is not clear to me. May be few lines should be added to justify the work. 2// Authors have discussed linear and non-linear models in section 2.3 (section-1). It is mentioned that the non-linear model is out of the scope of the present work. However, the motivation to use linear model over non-linear model is missing. Please discuss the reason to prefer linear model alone. 3// By looking at the Figures 6, 7 and 8, the results are almost similar. Same is mentioned in the conclusion section (286-289). Authors mentioned that the substrate RO4350B is matching better to CubeSat dimension requirements. Will it be possible for authors to compare these results for already existing results, for example, for other substrates mentioned in Point 1 above and also in the introduction section of the manuscript. 4// Upon comparing only the dimensionality and stating it to be the best, may not be an ideal case for practical use. Authors should also discuss the negative side of substrate RO4350B, so that the young minds can be motivated to pursue research work in that direction. Based on these observations, I recommend MINOR revision for the present manuscript.Author Response
Thank you for your comments. Please see the attachment.

Reviewer 2 Report
An interesting paper entitled “Signal conditioning stage in S-band communication subsystem for CubeSat applications.”
The abstract and the conclusion contain some appreciable quantitative and qualitative performance metrics.
The paper would benefit from an inclusion of detailed design and analysis of the transmitter front-end subsystem performance metrics including linearity and efficiency for the considered substrates. Moreover, state the signal conditioning PA’s stability response(s) for the considered substrates up to the cut-off frequency to eliminate potential fabrication defects. How would your choice of substrate design cater for applications that have a high linearity requirement over power-added efficiency? The presented design should be tested with real world modulated signals to ascertain its use case application performance metrics.
Furthermore, the authors should provide the spectrum for the output of the main power amplifier (PA) with a bandpass filter after it. They should compare this spectrum with their presented design (with no bandpass filter before the antenna). A detailed analysis of the PA’s power-added efficiency and the gain compression point should be frontally presented.
Author Response
We really appreciate your suggestions. Please see the attachment.

Reviewer 3 Report
I believe the authors have done significant work in this project through design, simulation and fabrication. However, I do not think this paper is organized in an academic publication but seems to be a product development summary. For instances:
- I understand the three different substrates are the foundation of the proposed innovation, but the reason why the can bring the performance gain is not illustrated clearly. It supposes to be explained in theory and compared to the legacy methods in details.
- The architecture design in section 2.2 also needs to be compared to the legacy designs.
- The performance differences observed in the physical dimensions need to be further analyzed.
- The academic contribution of this paper is unclear. It supposes to be summarized with a more solid related work section.
- Any disadvantages of this new design?
Round 2
Reviewer 2 Report
The authors have not holistically addressed the following comments thus:
The paper would benefit from an inclusion of detailed design and analysis of the transmitter front-end subsystem performance metrics including linearity and efficiency for the considered substrates. Moreover, state the signal conditioning PA’s stability response(s) for the considered substrates up to the cut-off frequency to eliminate potential fabrication defects. How would your choice of substrate design cater for applications that have a high linearity requirement over power-added efficiency? The presented design should be tested with real world modulated signals to ascertain its use case application performance metrics.
Furthermore, the authors should provide the spectrum for the output of the main power amplifier (PA) with a bandpass filter after it. They should compare this spectrum with their presented design (with no bandpass filter before the antenna). A detailed analysis of the PA’s power-added efficiency and the gain compression point should be frontally presented.
A compelling response is required please.
Author Response
The authors appreciate the reviewer enriching comments and suggestions, Please see the attachment

Reviewer 3 Report
I believe the authors have addressed my comments properly. Thus, I have no further concern for the publication. It will be better if the response to my 2nd comment can be included in the paper briefly.
Thanks for the efforts of the authors!
Author Response
Thank you for your comments, please see the attachment

Round 3
Reviewer 2 Report
It is appreciable to have a future work related to the presented research findings provided the fundamental research context is judiciously addressed.
Most relevant research questions bordering on this paper are unaddressed by the authors' work or through a literature review.
"The paper would benefit from an inclusion of detailed design and analysis of the transmitter front-end subsystem performance metrics including linearity and efficiency for the considered substrates. Moreover, state the signal conditioning PA’s stability response(s) for the considered substrates up to the cut-off frequency to eliminate potential fabrication defects. How would your choice of substrate design cater for applications that have a high linearity requirement over power-added efficiency? The presented design should be tested with real world modulated signals to ascertain its use case application performance metrics. Furthermore, the authors should provide the spectrum for the output of the main power amplifier (PA) with a bandpass filter after it. They should compare this spectrum with their presented design (with no bandpass filter before the antenna). A detailed analysis of the PA’s power-added efficiency and the gain compression point should be frontally presented. A compelling response is required please."
Author Response
Thank you very much for your comments, Please see the attachment
